# Reproducing "Identifying through flows for recovering latent representations"

## 1 Reproducibility Summary

### 2 Scope of Reproducibility

3 The authors claim to introduce a model for recovering the latent representation of observed data, that outperforms
4 the state-of-the-art method for this task; namely the iVAE model. They claim that iFlow outperforms iVAE both in
5 preservation of the original geometry of source manifold and correlation per dimension of the latent space.

### 6 Methodology

7 To reproduce the results of the paper, the main experiments are reproduced and the figures are recreated. To do so, we
8 largely worked with code from the repository belonging to the original paper. We added plotting functionality as well as
9 various bug fixes and optimisations. Additionally, attempts were made to improve the iVAE by making it more complex
10 and fixing a mistake in its implementation. We also tried to investigate possible correlation between the structure
11 of the dataset and the performance. All code used is publicly available at `https://github.com/HiddeLekanne/`
12 `Reproducibility-Challenge-iFlow`.

### 13 Results

14 The obtained mean and standard deviation of the MCC over 100 seeds are within 1 percent of the results reported in
15 the paper. The iFlow model obtained a mean MCC score of 0.718 (0.067). Efforts to improve and correct the baseline
16 increased the mean MCC score from 0.483 (0.059) to 0.556 (0.061). The performance, however, remains worse than the
17 performance of iFlow, further supporting the authors' claim that the iFlow implementation is correct and more effective
18 than iVAE.

### 19 What was easy

20 The GitHub repository associated with the paper provided most necessary code and ran with only minor changes. The
21 code included all model implementations and data generation. The script that was used to obtain results was provided,
22 which allowed us to determine which exact hyperparameters were used with experiments on the iFlow models. Overall,
23 the code was well organised and the structure was easy to follow.

### 24 What was difficult

25 The specific versions of the Python libraries used were unknown, which made it infeasible to achieve the exact results
26 from the paper when running on the same seeds. The code used to create figures 1-3 in the original paper was missing
27 and had to be recreated. Furthermore, the long time the models needed to train made experimentation with e.g., different
28 hyperparameters challenging. Finally, the code was largely undocumented.

### 29 Communication with original authors

30 Communication with the authors was attempted but could not be established.

# 1 Introduction

Nowadays, different types of deep generative models excel at generating new data by either explicitly or implicitly modelling the distribution of the training data. However, sometimes it is useful to recover the distribution that generated the observed data, i.e. the latent distribution, rather than the data distribution itself. It is easy to see that this is a more difficult task due to the unknown relation between the unobserved latent variables and the observed data. The concept of recovering the true latent distribution underlying the data is a form of *identifiability*.

Some research has been done in this area. Previously, models (notably $\beta$-VAE [1] and its variations) were created with the purpose of creating *disentangled* representations, where single latent units correspond to single generative factors. While related to identifiability, such models do not provide any proof or guarantee that they can recover the true latent representations.

More recently, an identifiable variation of the VAE called iVAE was proposed [4], which uses a factorised prior conditioned on an auxiliary variable to guarantee a basic form of identifiability. In practice however, the fact that this model optimises a lower bound on the posterior, rather than the actual posterior, could negatively affect the capability of the model to recover the true latent variables.

The paper "Identifying through flows for recovering latent representations" proposes *iFlow*, a model that aims to alleviate these problems by using Normalising Flow models rather than VAEs [8]. The fact that Normalising flows model exact distributions rather than approximating the posterior could make them more suitable for this task.

# 2 Scope of reproducibility

In this review, the work of the proposed iFlow model by Li et al. [8] is reproduced and examined. The aim is to reproduce the results obtained by the authors and to investigate the claims made in the paper. The claims made can be seen below. Each claim will be examined in a corresponding subsection in section 4.

1. Simulations on synthetic data validate the correctness and effectiveness of the proposed iFlow method and demonstrate its practical advantages over other existing methods.

2. iFlow outperforms iVAE in identifying the original sources while preserving the original geometry of source manifold.

3. iFlow exhibits much stronger correlation than iVAE does in each single dimension of the latent space.

4. Making iVAE more expressive does not help it approximate the real latent space further, justifying the discrepancy in parameters.

# 3 Methodology

Most of the original source code was available and used to test the reproducibility of the paper which can be found in the corresponding GitHub repository [1]. This repository itself contained code from the repository of the iVAE model[2] and *nflows* [3].

The iFlow implementations were used largely as is, while the iVAE implementation was refactored to apply modifications more easily. The *nflows* code base has been removed from the repository and imported as a library instead. Furthermore, some small optimisations were made to make certain functions more efficient by vectorising them. For reproducing the results the models were trained on a GPU (see section 3.5). The code for creating the visualisations was not included in the repository and was therefore recreated. The implementation was made using the `PyTorch` and `NumPy` libraries with `Python 3.7.9`. `TensorBoard` was used for logging of variables during training.

## 3.1 Model descriptions

The paper compares two models: the proposed iFlow model and the iVAE model.

The iFlow model is a variation on the Normalising Flow model *rational-quadratic neural spline flows (featuring autoregressive layers)* (RQ-NSF (AR)) [2], where the prior has been replaced with a factorised exponential prior

---

[1]`https://github.com/MathsXDC/iFlow`

[2]`https://github.com/siamakz/iVAE`

[3]`https://github.com/bayesiains/nflows`

distribution conditioning the latent variables $z$ on auxiliary variables $u$ to obtain identifiability up to an equivalence relation. The natural parameters of the prior are obtained through a trainable multi-layer perceptron (MLP) which takes the auxiliary variable $u$ as input. Each iFlow model contains approximately 3 million trainable parameters.

The iVAE model is implemented as an extension of vanilla VAE models [6], using MLPs for both the encoder and decoder. The number of layers, hidden dimensions and activation functions are hyperparameters. The encoder uses two MLPs (one for mean and variance each), while the decoder uses just one. Additionally, the prior mapping the auxiliary variables $u$ to the latent variables $z$ is also implemented as an MLP. In total, the iVAE model with standard parameters has roughly 18,000 trainable parameters.

There is a significant difference in the complexities of iFlow and iVAE, seen in the number of trainable parameters the models have. The authors argue that this is not the cause of the inferior performance of the iVAE, showing that adding more layers/increasing the hidden dimensions of the model does not increase performance. However, only a limited range of parameters were used for this, resulting in only weak evidence to support the claim that the comparison is fair. We further investigate this claim by scaling up the complexity of iVAE through various methods, namely adding residual connections and layer normalisation in addition to changing hyperparameters.

When looking at the implementation of the iVAE model, there appears to be a difference with the theory: the mean of the prior distribution is not a function of the auxiliary variables $u$, as the theory states, but simply fixed to be 0 at all times. We aim to incorporate this change into the implementation of iVAE to see if it leads to better performance.

## 3.2 Dataset

A synthetic dataset is required in order to truly know the underlying latent distribution, which is necessary for quantitative analysis of the performance. The authors chose to use a dataset consisting of sources of non-stationary Gaussian time-series. Such data was previously used to introduce time-contrastive learning as a means of achieving identifiable non-linear independent component analysis [3] and was additionally used to assess the performance of the iVAE [4].

The latent representation (source) is created as non-stationary Gaussian time series. This data consists of $M$ segments, which are modelled as Gaussian distributions with different, randomly selected mean and variance. The means are sampled from uniform distribution $[-5, 5]$, while the variances are sampled from uniform distribution $[0.5, 3]$. Each segment contains $L$ samples drawn from the corresponding distribution of segment $M$. The segment labels serve as the auxiliary variables $u$.

A 3 layer invertible MLP is used to transform the samples in a non-linear manner to obtain the observable data. The invertible MLP consists of mixing matrices with the non-linear activation function $h(x) = tanh(x) + \alpha \cdot x$. The last layer does not contain a non-linear activation function. Due to the constraints of Flow models, the dimensionality $d$ of these observed data points has to be the same as the dimensionality of the latent representation $n$. The data generator allows for the addition of noise to the data points, but this is not utilised.

In the paper, results are reported on a dataset created using $M = 40$, $L = 1000$, $n = d = 5$ and $\alpha = 0.1$. For visualisations of the sources and the estimations of the models, $M = 5$, $L = 1000$, $n = d = 2$ and $\alpha = 0.1$ are used. This differs from the reported $M = 40$ from the original paper where the figure indicates that the true $M = 5$.

## 3.3 Hyperparameters

The authors of the original paper mention specific values for some of the hyperparameters. However, for other hyperparameters only a range is provided without a clear indication of what values were used for each evaluation.

As mentioned before, for generating the data, the parameters $M = 40$, $L = 1000$, $n = d = 5$ and $M = 5$, $L = 1000$, $n = d = 2$ were used for experiments and visualisation respectively. A factorised Gaussian distribution was used as a prior for the source distributions. The means and variances for these distributions were sampled from uniform distributions $[-5, 5]$ and $[0.5, 3]$ respectively. The data was transformed with an invertible MLP of depth 3 with `tanh` activation function and a slope of $0.1$.

Both the iVAE and iFlow used the same batch size ($B = 64$) and learning rate of $0.001$. A learning rate drop factor of $0.25$ was used and a learning patience of 10. An Adam optimiser without weight decay and with standard $\beta$ values (0.9, 0.999) and $\epsilon$ (1e-8) was used [5]. A learning rate scheduler to reduce the learning rate with a factor 0.1 on plateaus ensured that the learning rate decreased over time.

The iFlow models were initialised with a flow length of 10 with 8 bins. The Rational Quadratic Neural Spline Flows with Autoregressive transforms (RQNSF-AR) was used as flow type. The Softplus activation was exerted on the natural parameters.

To replicate the iVAE baseline, a model with a hidden dimensionality of 50, a latent dimension equal to that of the data ($d = n = 5$) and 3 layer MLPs with leaky ReLU with $\alpha = 0.1$ as activation function. These same hyperparameters were used for the additional experiments.

## 3.4 Experimental setup and code

The code for this reproducibility review is publicly available at `https://github.com/HiddeLekanne/Reproducibility-Challenge-iFlow`. As mentioned earlier, this code consist of a combination of the iFlow and iVAE codebases (see section 3).

The iFlow and iVAE models were trained with 100 different seeds to generate datasets and the aforementioned hyperparameters. As is standard for these types of models, the iFlow model was trained using negative log likelihood as a loss, and the iVAE was used using the ELBO as a loss [6]. Model performance was evaluated using the mean correlation coefficient (MCC) between the original source of the data and the estimated latent variables from the models.

## 3.5 Computational requirements

All experiments were run on the LISA system[4] provided by the University of Amsterdam. This system provides multi-core nodes for research projects. A GeForce GTX 1080 Ti was used to train the models.

Training of iVAE models for 100 different seeds took approximately 2 hours. Training of a single iFlow model with a flow length of 10 took approximately 45 minutes. To alleviate some of the computational cost, the 100 models were trained in two worker nodes instead of one. This totalled to approximately 1.5 days of training per 100 models.

Upgrading of computational resources would not garner better results, with respect to time, based on the fact that the bottleneck for the computations was the speed of a single thread CPU. Attempts were made to improve this performance but these did not decrease training time.

## 4 Results

In this section, results from the original paper are recreated. In addition, some further experiments were done of which the results can also be found below. These additional experiments consist of improving the existing base line proposed in the paper, as well as exploring the relation between the complexity of the synthetic data and the achieved MCC scores.

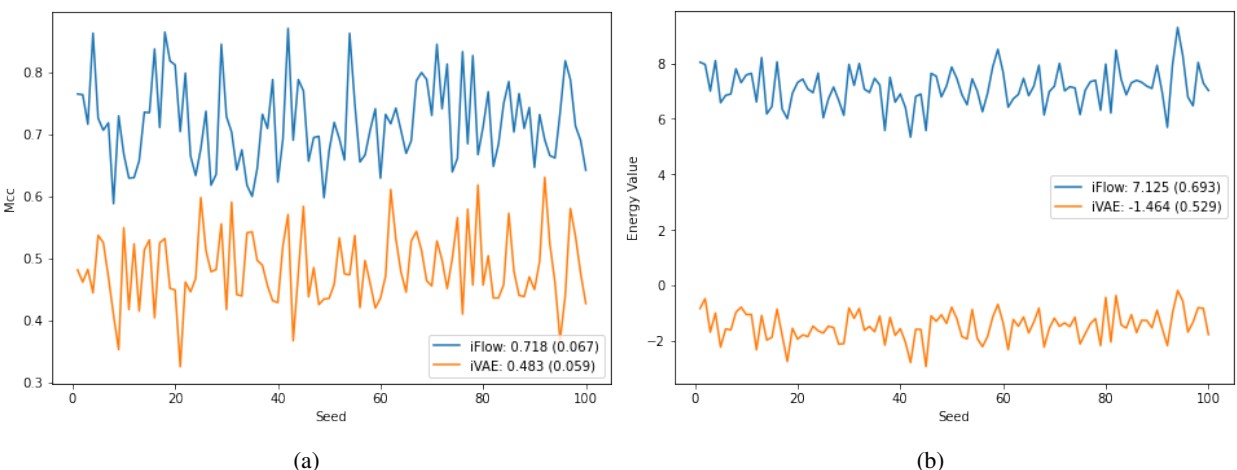

|        |        |
|--------|--------|
| (a)    | (b)    |

Figure 1: Comparison of identifying performance (MCC) and the energy value (log-likelihood) versus seed number respectively.

---

[4]`https://userinfo.surfsara.nl/systems/lisa`

### 4.1 Results reproducing original paper

#### 4.1.1 Comparison of identifying performance

The MCC scores and log-likelihood over 100 seeds are displayed in figure 1a and figure 1b respectively. The figures show that there is high variance in MCC scores for different datasets. The iFlow models obtained a mean accuracy of $0.718$ with a standard deviation of $0.067$ whereas the iVAE models obtained a mean accuracy of $0.483$ with a standard deviation of $0.059$ which is in compliance with the results produced in the original paper. The results for the iVAE models are significantly worse than in the original iVAE paper. An improvement to the implementation was made to better emulate the performance of this paper which resulted in a fairer comparison (see section 4.2.1).

As can be seen in figure 1b, the energy values of the iVAE are significantly lower compared to those of iFlow, matching the results of the paper. The authors noted that the difference in energy values could indicate that the gap between the ELBO and the actual log likelihood is not negligible.

#### 4.1.2 Preservation of original source manifold geometry

Figure 2 shows the 2D visualisation for different data seeds. The original paper stated that an $M = 40$ was used but figures indicated that this should be $M = 5$.

The results largely support the claim of the author that the original geometry of the source manifold is preserved. The estimations from the iFlow model seem more similar to the original source than the estimations from the iVAE models, although it still contain artefacts from the observations. Figure 2a is an example of such where the latent dimensions are not successfully recovered. In other examples, the original Gaussian distributions are mostly recovered apart from some transformation as was the case in the original paper. The collapse of the latent space from iVAE models observed in the original paper was not prevalent during experiments. However, the preservation of the original geometry of the source manifold is better captured by the iFlow models.

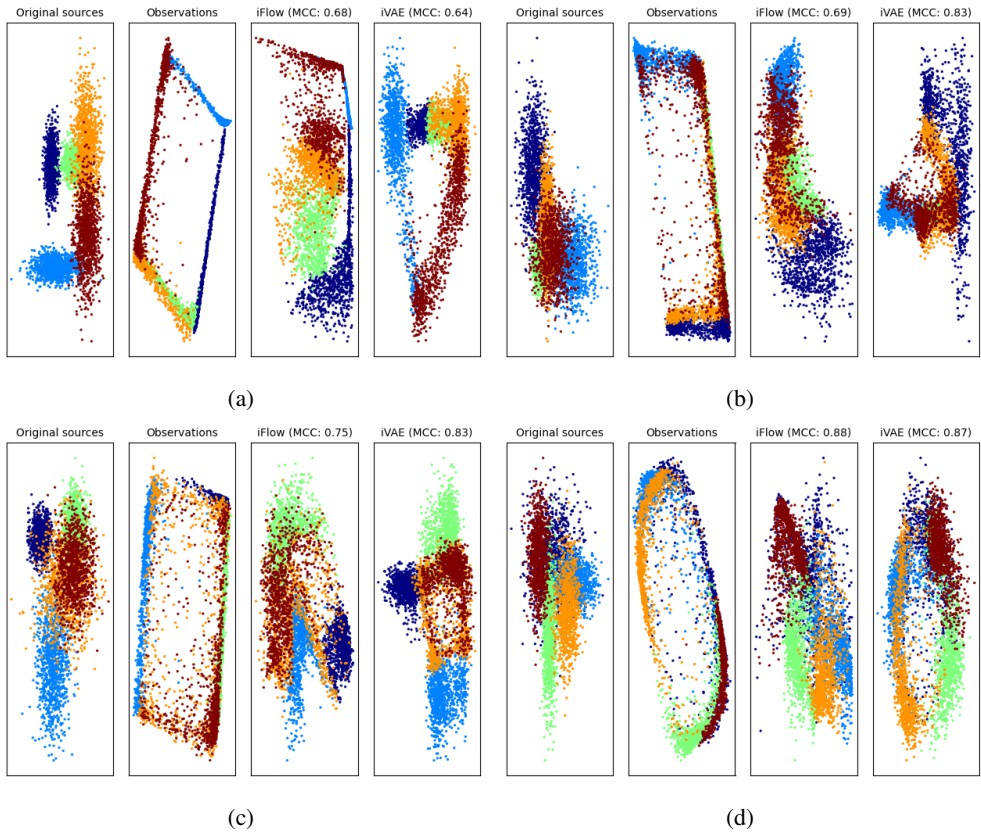

Figure 2: Visualisation of 2D-cases

### 4.1.3 Separate latent dimension correlation

Figure 3 and figure 4 show the correlation between the source signal used to generate the data and the latent variables recovered by the iFlow and iVAE models. Figure 3 shows the results of the best performing iFlow, which we assume is what figure 3 of the original paper also depicts. For fairness, we also show the results for the dataset that iVAE performed best on in figure 4.

These results largely support the claim that iFlow exhibits stronger correlation than does iVAE in each single dimension of the latent space: while this is generally the case, it does occur for some datasets that iVAE has a higher correlation coefficient than iFlow on one or even two of the latent dimensions, as shown in figure 4.

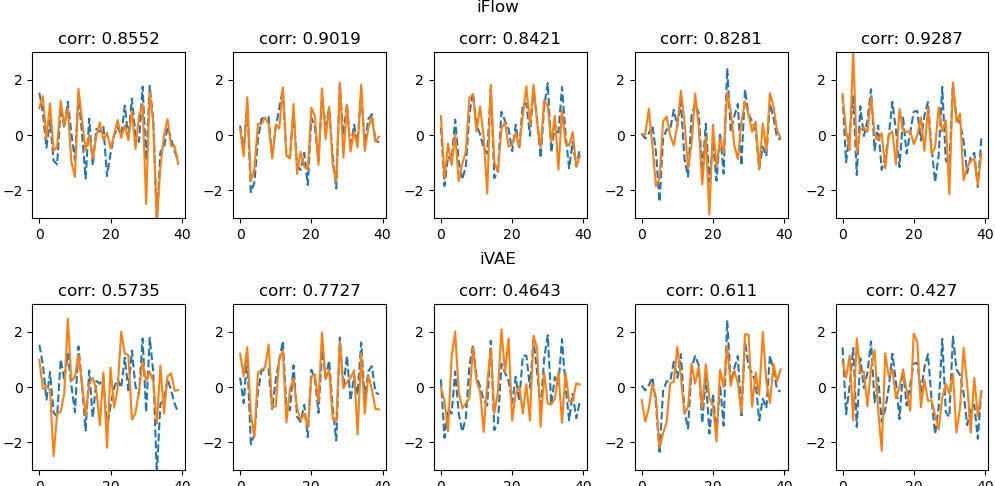

Figure 3: Comparison of the latent variables recovered by the models (orange lines) to the true latent variables (dashed blue lines) for individual dimensions. This figure shows results for the seed that resulted in the best iFlow performance.

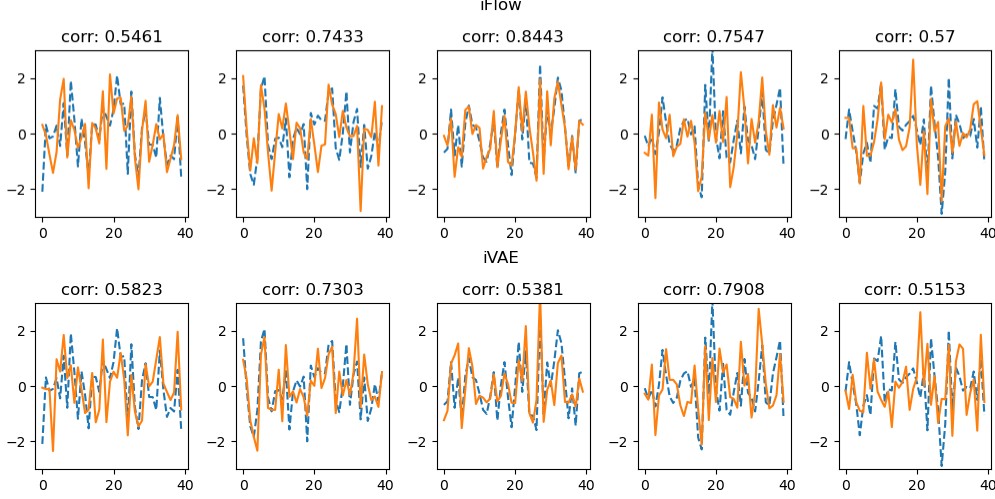

Figure 4: Comparison of the latent variables recovered by the models (orange lines) to the true latent variables (dashed blue lines) for individual dimensions. This figure shows results for the seed that resulted in the best iVAE performance.

### 4.2 Results beyond original paper

#### 4.2.1 Improved baseline

In figure 5a and 5b the MCC scores and energy values over 100 seeds are displayed for the iFlow model, iVAE model and improved iVAE model. The addition of the trainable mean, based on auxiliary parameters, shows an increase in the mean MCC score from 0.483 (0.059) to 0.556 (0.061). The ELBO score improves almost with a constant value for every seed.

Other attempts had been made to improve this baseline by increasing the complexity of the iVAE model. However, were tried before the mistake in the iVAE implementation had been noticed, and are therefore not very useful. These results can be seen in appendix B.

In addition to figure 1, recreations of figures 2 3 using the improved baseline were made. These are not included in this report but can be found in appendix C

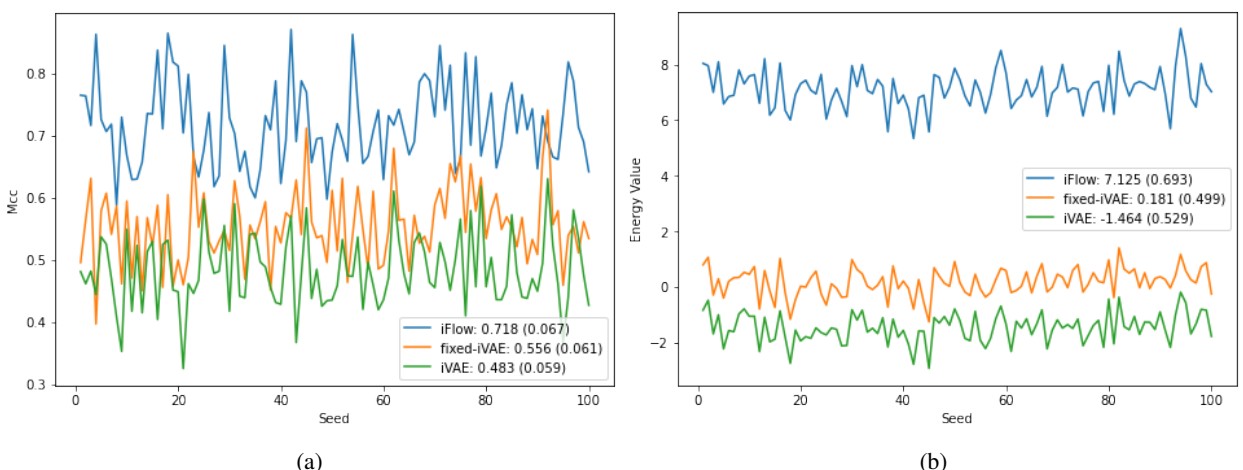

(a) (b)

Figure 5: Comparison of identifying performance (MCC) and the energy value (log-likelihood) versus seed number respectively, with the fixed version iVAE included.

#### 4.2.2 Synthetic data complexity

To measure the complexity of the dataset, the mean Kullback-Leibler divergence [7] of each source and its nearest neighbour was used.

This metric showed no correlation (0.06) with the MCC scores obtained by iVAE or iFlow, showing that the randomly sampled parameters of the source distribution were likely not to blame for the high variance in MCC scores.

## 5 Discussion

The results reproduced in the previous section largely support the claims by the original authors. Firstly, the MCC scores that we obtained after training the model on synthetic data are very similar to the ones reported. Secondly, the recreated visualisation of 2D latent sources seems to support the claim that the iFlow method outperforms iVAE in identifying the original sources. Finally, the claim that iFlow exhibits much stronger correlation than iVAE in each single dimension of the latent space is not fully supported by our results. In the original paper, the authors show this correlation only for the best iFlow results. When visualising the individual dimensions of the latent variables for the best iVAE results, iVAE outperforms iFlow in two of the latent dimensions. This shows that the claim does not strictly hold true for all seeds. Nevertheless, since iFlow outperforms even the best iVAE on most latent dimensions, it still seems to be a reasonable claim.

Our experiments to improve the performance of the iVAE model, by modelling the prior means as a trainable function of the auxiliary variable $u$, managed to increase its performance significantly. However, the performance remains worse than that of iFlow. This further cements the claim that the iFlow model is more suited for the task of identifiability than iVAE.

The strength of our approach was that we were generally faithful to the original implementation, using largely the same code which we examined thoroughly. Therefore, the chance of implementation differences with the original code is very small. Additionally, we rigorously compared the code with the underlying theory, allowing us to correct an important mistake in the baseline.

A weakness of our approach was that we did not do any work to examine the models on a more realistic dataset, meaning the generalisability of the model remains an open question. Furthermore, due to the high variance in the results of identifying models, all experiments had to be run with a large number of seeds (100), which took a long time given the fact that training of a single model took approximately 40 minutes. For this reason, experimentation done with hyperparameters was limited. The experiments in the appendix of the paper were not replicated for similar reasons. These experiments looked at the effect of different activation functions on the performance of iFlow and the effect of more and larger hidden layers on the performance of iVAE.

Overall, the authors provided a model which outperforms the previously best method for this problem in a quantifiable measure. Additionally, high variance in the results is addressed appropriately by running the experiments over a large number of seeds. Furthermore, the visualisation of the true sources and the estimations by the models makes it easier to interpret the MCC scores. Lastly, the model is theoretically well motivated.

Despite these strengths of the original paper, some improvements could be made to further substantiate the claims made in the paper. There is a clear advantage that iVAE has over iFlow, which is not mentioned by the authors: iVAE can be used when the dimensionality of the latent sources differs from the data dimensionality, while iFlow cannot. The fact that iFlow needs data with such corresponding dimensionalities also means that the iVAE had to be trained without a bottleneck. This is an important part of the VAE architecture, and the lack thereof could have contributed to the weaker performance of iVAE; compared to the paper introducing iVAE (MCC of above 0.95), the MCC scores of the iVAE reported by the authors are significantly worse (MCC of 0.496). This discrepancy is not addressed or explained by the authors.

## 5.1 What was easy

The code provided in the GitHub repository worked almost out of the box, with only small adjustments needed; the source code of the nflows library that was included in the repository was replaced with an import. This fixed an issue that prevented the code from running on a CPU. The code was well organised into separate files for e.g., the iFlow model, iVAE models or training, making it easy to quickly find specific parts of the code when needed. The code that generates the data the models are trained on also came with the implementation, and worked without any issues.

With the code, a shell script was provided that seems to be the one used for the experiments on iFlow in the paper (although this was not explicitly stated). This allowed for easy replication of these experiments, with all of the used hyperparameters provided.

## 5.2 What was difficult

There were difficulties in replicating some parts of the paper. The lack of a provided environment means that our code was likely run using different versions of some libraries such as PyTorch or NumPy. This could have contributed to the difference in outcomes of our experiments compared to the paper while using the same seeds.

While the script used to run iFlow experiments was provided, the same was not true for the iVAE experiments. This was not a large problem, however, since the authors do state that the hyperparameters used are the same as in the original iVAE paper [4]. The code for creating plots (Figures 1,2,3 in the iFlow paper) was also not provided and additional code had to be written to recreate these figures.

The training of the iFlow models for all 100 seeds took a significant amount of time. With the training for one seed taking approximately 40 minutes, the full training took roughly a day and a half (running two batches of 50 seeds simultaneously). This made it difficult to do full-scale experiments with different hyperparameters.

Lastly, there was a large portion of unused code present in the repository, which made it more difficult to understand the overall structure of the code. This includes the source code of the nflows library, code for planar flows, multiple different iVAE variations, an alternative dataloader, an unused dataset and an implementation of training using annealing.

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

## A  Alternative iFlow model

A comment in the files mentioned an error in the implementation because the Softplus function was applied to $\xi$ as well as $\eta$ from the natural parameters $\lambda(\mathbf{u})$. An alternative version of the implementation was also tested where the Softplus activation function was only exerted on $\xi$, as there are no constraints on the sign of $\eta$.

The results obtained using this method were approximately the same as the original performance of the iFlow. A mean MCC of 0.72 with a standard deviation of 0.057 was achieved. Because these results were not a significant improvement, it was decided to include this experiment as an appendix.

## B  Baseline improvement experiments

The table below shows the results of experiments with changing the iVAE architecture to increase complexity. As shown, adding skip connections or layer normalisation to the architecture did not increase performance with respect to the unchanged baseline. Due to the high training time, no additional experiments could be done.

| addition | NUM_HIDDEN | NUM_LAYERS | AVG MCC |
|---|---|---|---|
| - | 50 | 3 | 0.483 (±0.059) |
| residual connections | 50 | 3 | 0.474 (±0.053) |
| layer normalisation | 50 | 3 | 0.461 (±0.051) |

 # C   Visualizations for Fixed iVAE

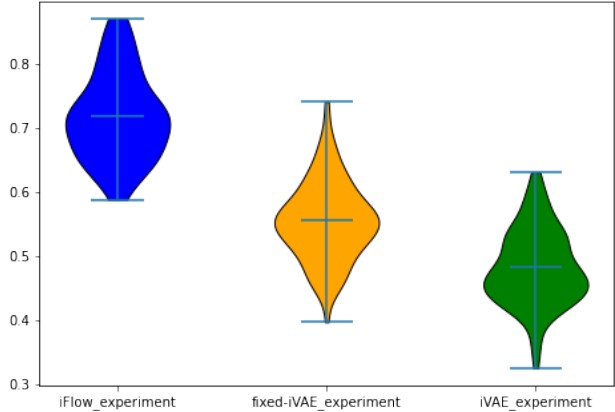

Figure 6: Alternative visualisation of the MCC scores obtained by the models, including the fixed iVAE.

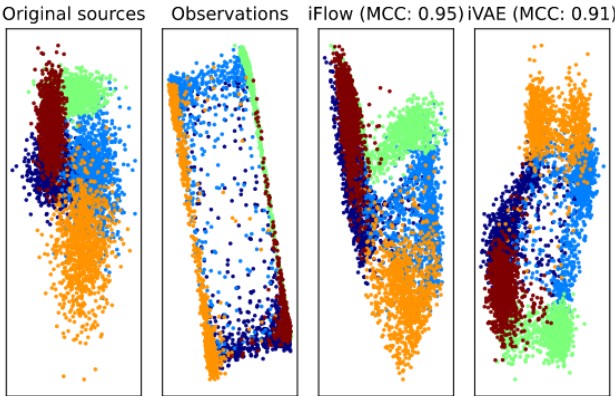

Figure 7: Visualisation of 2D-cases, comparing iFlow to the fixed version of iVAE.

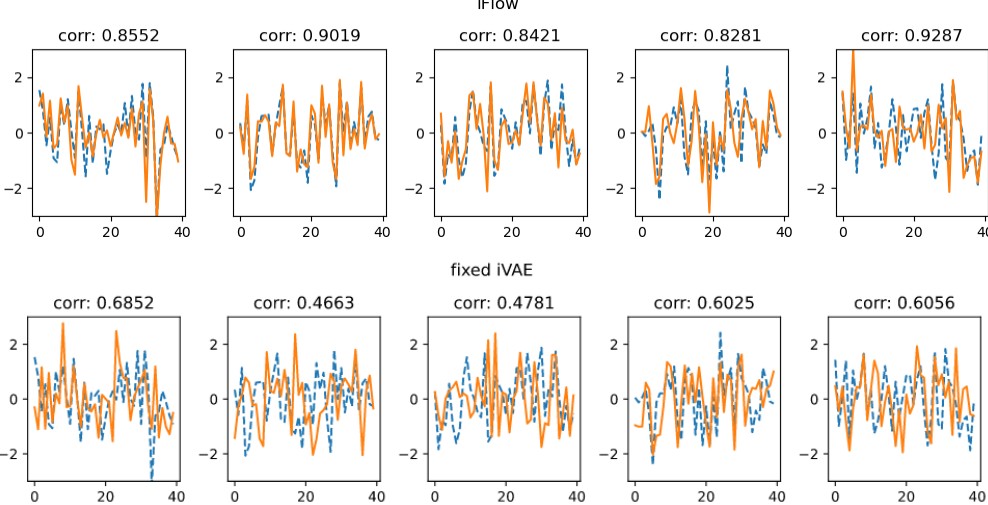

Figure 8: Comparison of the latent variables recovered by the models (orange lines) to the true latent variables (dashed blue lines) for individual dimensions. This figure shows results for the seed that resulted in the best iFlow performance.

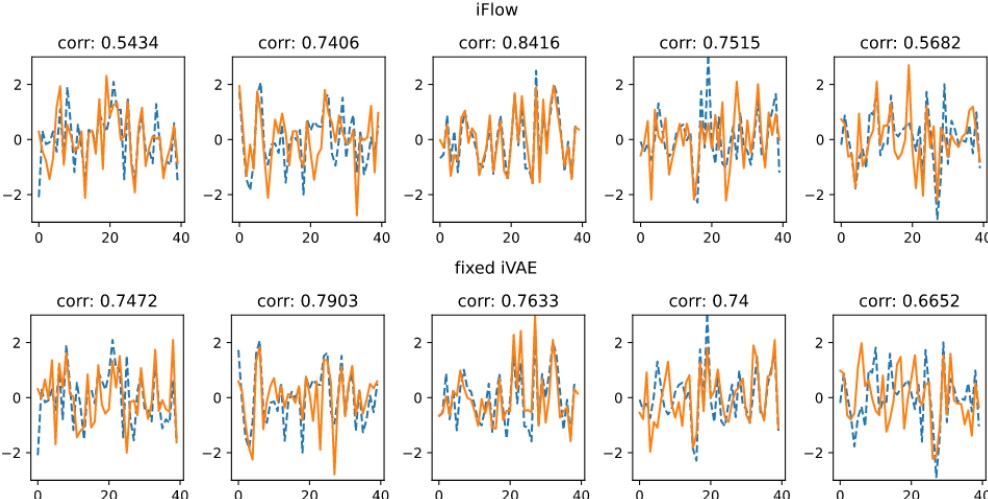

Figure 9: Comparison of the latent variables recovered by the models (orange lines) to the true latent variables (dashed blue lines) for individual dimensions. This figure shows results for the seed that resulted in the best fixed iVAE performance.

