# OpenReview forum: "Reproducing "Identifying through flows for recovering latent representations""
_ML_Reproducibility_Challenge/2020 — RC2020_

### Official Review · AnonReviewer1 · 2021-02-28
**A clear and fair reproductivity assessment report**

**Rating:** 7
**Confidence:** 3

**Review:**

This report confirms most of the claims in the original paper and points out some minor issues of the proposed iFlow method that are not fully supported by the experimental observations. Furthermore, the report also mentions that the MCC scores of the iVAE
reported by the authors are significantly worse than those in the iVAE paper. The limit of the report is the lack of experimental evaluations traversing different hyperparameter values due to the computation bottleneck.  The report states the coverage limit clearly.

**Familiar With The Original Paper:**

I have read the original paper

**Reproducibility Summary:**

Report has summary

---

### Official Review · AnonReviewer3 · 2021-03-03
**Review for reproducibility challenge**

**Rating:** 7
**Confidence:** 4

**Review:**

The overall paper presents good ablation with a clear introduction to the problem statement in the reproduction summary.

It was good to see a detailed hyper-parameter search over the baseline model.
But, it would have been great if the authors would have extended the work over real datasets in support of the points mentioned in the discussion section.
Also, if the authors can share more details regarding the time complexity for running the experiments which would help future researchers to tackle this problem first.
The discussion section mentioned by the authors listing down the strengths and weakness of the method which is a great reference point for future work.
From the paper, it seems that the authors may not have any direct communications with the original authors. They mainly obtained information from the original paper and the original codebase.
The authors have clearly stated the scope of reproducibility with clarity in the claims they learned from the original paper with further sections explaining the clams.
Small suggestion: In page 9: section-"Baseline improvement experiments" if the table can be written in a more representable manner.



**Familiar With The Original Paper:**

I have read the original paper

**Reproducibility Summary:**

Report has summary

---

### Decision · Program_Chairs · 2021-03-31

**Decision:**

Accept

**Comment:**

A very thorougj reproduction, even extending the results of the original paper.